# Anti-Schmallenberg Virus Activities of Type I/III Interferons-Induced Mx1 GTPases from Different Mammalian Species

**DOI:** 10.3390/v15051055

**Published:** 2023-04-25

**Authors:** Calixte Bayrou, Anne-Sophie Van Laere, Phai Dam Van, Nassim Moula, Mutien-Marie Garigliany, Daniel Desmecht

**Affiliations:** 1Animal Pathology, FARAH Research Center, Faculty of Veterinary Medicine, University of Liège, Sart-Tilman B43, 4000 Liège, Belgium; calixte.bayrou@uliege.be (C.B.); as.vanlaere@uliege.be (A.-S.V.L.); dvphai@vnua.edu.vn (P.D.V.); mmgarigliany@uliege.be (M.-M.G.); 2Animal Productions, FARAH Research Center, Faculty of Veterinary Medicine, University of Liège, Sart-Tilman B43, 4000 Liège, Belgium; nassim.moula@uliege.be

**Keywords:** Mx protein, interferon, innate immunity, Schmallenberg virus, orthobunyaviruses

## Abstract

Mx proteins are key factors of the innate intracellular defense mechanisms that act against viruses induced by type I/III interferons. The family Peribunyaviridae includes many viruses of veterinary importance, either because infection results in clinical disease or because animals serve as reservoirs for arthropod vectors. According to the evolutionary arms race hypothesis, evolutionary pressures should have led to the selection of the most appropriate Mx1 antiviral isoforms to resist these infections. Although human, mouse, bat, rat, and cotton rat Mx isoforms have been shown to inhibit different members of the Peribunyaviridae, the possible antiviral function of the Mx isoforms from domestic animals against bunyaviral infections has, to our knowledge, never been studied. Herein, we investigated the anti-Schmallenberg virus activity of bovine, canine, equine, and porcine Mx1 proteins. We concluded that Mx1 has a strong, dose-dependent anti-Schmallenberg activity in these four mammalian species.

## 1. Introduction

Type I (IFN-α/β) and III (IFN-λ) interferons provide powerful and universal innate intracellular defense mechanisms against viruses [1]. Among the induced antiviral effectors, the Mx proteins of some species appear as key components of defense. The Mx structure resembles that of other members of the dynamin-like large GTPase superfamily, consisting of an N-terminal GTPase domain (G domain) and a C-terminal stalk [2]. These two structural domains are linked by a bundle-signaling element that is necessary to transfer structural changes during GTP binding and hydrolysis to the stalk structure.

In 1962, Lindenmann showed that the inbred mouse strain A2G is resistant to doses of mouse-adapted influenza virus that are lethal to other inbred strains [3]. The presence of a natural resistance gene to influenza was intriguing because mice are not natural hosts for orthomyxoviruses. It soon became clear that the concerned gene (*mx1*) was the first member of a small gene family present in all vertebrate species from fish to humans and that the spectrum of antiviral activity was much larger than initially appreciated. Today, we know that Mx family members have distinct antiviral profiles against a diverse range of viruses, including pathogens of great importance in human and veterinary medicine, as summarized in [4,5].

Typically, viruses susceptible to Mx inhibition have a genome made of single-stranded RNA strands of negative polarity. The two viral families by far the most studied in this regard are *Orthomyxoviridae* and *Peribunyaviridae*. The family *Bunyaviridae* was formerly the largest virus family, with more than 350 viral species encompassed by five genera: *Orthobunyavirus*, *Hantavirus*, *Nairovirus*, *Phlebovirus*, and *Tospovirus* [6,7]. In 2016, the International Committee on Taxonomy of Viruses decided to elevate the family *Bunyaviridae* to the level of an order, designated *Bunyavirales*. This new order contains at least 11 families and 15 genera of bunyaviruses [8]. Many of them are significant pathogens of humans, animals, or plants.

Human illnesses resulting from bunyavirus infection range from mild, asymptomatic infection to more severe diseases, including pulmonary diseases, hemorrhagic fevers, and fatal encephalitides [9,10,11,12]. For example, Crimean–Congo hemorrhagic fever virus (CCHFV, *Nairovirus*) is the second most widespread of all medically important arboviruses after dengue viruses and is endemic in much of Africa, Asia, and Europe. Notably, mortality rates of up to 30% in humans have been reported for CCHFV [13]. In addition, La Crosse virus (LACV, *Orthobunyavirus*) and Rift Valley fever virus (RVFV, *Phlebovirus*) are important causes of arboviral encephalitides. There are currently no effective therapeutics or vaccines for these human bunyaviral diseases. Major research efforts have thus concentrated on innate anti-bunyavirus effectors, including human and murine Mx1 proteins. This has led to the discovery that human MxA and mouse, rat, and cotton rat Mx2 proteins (but not their respective Mx1 isoforms) inhibit different members of the *Peribunyaviridae* [4]. Specifically, human MxA inhibits the growth of LACV, RVFV, CCHFV, Hantaan (HTNV), Puumala (PUUV), Tula (TULV), and Dugbe viruses in cell culture [14,15,16,17,18] and LACV in transgenic mice [19]. Mouse Mx2 slows HTNV replication [20], rat Mx2 inhibits LACV and RVFV [21], and cotton rat Mx2 decelerates RVFV replication [22]. In bats, it has been shown that Mx1 from *E. helvum* and *C. perspicillata* has anti-bunyavirus activities, whereas Mx1 from *P. pipistrellus* does not [23].

The family *Peribunyaviridae* also includes many viruses of veterinary importance, either because infection results in clinical disease or because animals serve as reservoirs for arthropod vectors. For example, RVFV affects ruminants, in whom it causes acute necrotizing hepatitis and/or abortion; Nairobi sheep disease virus infects small ruminants, in whom it causes hemorrhagic enteritis; and Akabane, Schmallenberg, and Cache Valley viruses infect ruminants, resulting in developmental abnormalities of the central nervous system and the appendicular apparatus. According to the evolutionary arms race hypothesis, evolutionary pressures should have led to the selection of the most appropriate Mx1 antiviral isoforms to resist these infections. Although many animal species are affected by bunyaviral infections, the possible antiviral function of the Mx1 isoforms they produce has, to our knowledge, never been studied. In the following study, the anti-bunyaviral functions of canine, bovine, porcine, and equine Mx1 proteins were compared by measuring the amount of viral nucleoprotein synthesized in vitro after the infection of a cell monolayer by Schmallenberg virus (SBV), the newcomer of the family, assigned to the genus *Orthobunyavirus*. SBV infects domestic ruminants and was first detected in 2011 at the German–Dutch border region. Thereafter, it rapidly spread throughout Europe, where it is now present with an enzootic status [24]. Recent reports have established its presence in Iran and Uganda, hence drastically enhancing its geographical distribution [25,26]. In adult ruminants, the infection is almost asymptomatic. However, after in utero infection, the virus induces the destruction of large portions of the central nervous system, leading to severe myo-skeletal malformations [27]. This arthrogryposis-hydranencephaly syndrome is responsible for dystocia and stillborn fetuses, with significant negative economic consequences. In Belgium, the estimation of losses for cattle herds was EUR 65–107 per cow [28]. Moreover, Tilston-Lunel and colleagues [29] showed that SBV can efficiently reassort with the Oropouche virus, a member of the *Orthobunyavirus* genus targeting humans, stressing a potential risk of a tragic outbreak in the human population. In recent years, SBV has become an important model virus to study other viruses of the Simbu serogroup and related orthobunyaviruses.

## 2. Materials and Methods

### 2.1. Experimental Design

The anti-SBV activity of four distinct mammalian Mx1 proteins was measured by comparing the number of viral nucleoprotein-positive cells 5 h after infection in a sample of 100,000 cells expected to contain both Mx1-positive and Mx1-negative cell subpopulations. The operating procedure ensured that all cells were exposed to the same scenario and reagents, since they were cultured, transfected, washed, infected, fixed, and stained together in the same well/tube. Finally, a V5 epitope flanked all the recombinant Mx1 proteins studied, which allowed an absolute standardization of the labeling of the different Mx1-positive cells to be identified via flow cytometry. 

For transfection, four identical expression plasmids were used. They all encoded a single polyprotein, consisting of an RFP protein in the N-terminal position, a 2A peptide, and a V5-Mx1 protein in the C-terminal position. The spontaneous splitting of the polyprotein into its 2 constitutive proteins was expected, with the nuclear localization of the RFP protein and the cytoplasmic localization of the V5-Mx1 protein. The recombinant V5-Mx1 proteins targeted were the canine (suffix “cf”, standing for *Canis familiaris*), equine (“ec”, *Equus caballus*), porcine (“ss”, *Sus scrofa*), and bovine (“bt”, *Bos taurus*) isoforms. The anti-SBV activity of these V5-Mx1 proteins was measured 5 h after cell monolayers were infected. Four independent triplicate repeats were performed. It is noteworthy that the reported results all come from experiments whose V5-Mx1 expression rates were comparable (between 40 and 60%).

#### 2.1.1. Cells

Human embryonic kidney cells (HEK-293T) were maintained at 37 °C under a humidified atmosphere of 5% CO_2_–95% air in DMEM supplemented with 10% fetal calf serum, 2 mM L-glutamine, 0.4 mM sodium pyruvate, 1× non-essential amino acids, 100 units/mL penicillin, and 0.1 mg/mL streptomycin. 

#### 2.1.2. Plasmids

A plasmid vector (pDA657RA) was synthesized for expressing a reference canine Mx1 (NP_001003134.1) in mammalian cells. The recombinant canine Mx1 (cfMx1) intended was fused to an N-term functional group consisting of a V5 epitope. The resulting V5-cfMx1 protein was further fused to an N-term bipartite group consisting of the fluorescent marker protein RFP followed by the spontaneously cleavable peptide 2A. The coding sequence (CDS), encoding the recombinant V5-cfMx1 protein, was codon-optimized, artificially synthesized via solid-phase DNA synthesis, and cloned in the aforementioned vector. Once the capacity of pDA657RA/V5-cfMx1 to drive the appropriate expression of V5-cfMx1 in HEK-293T cells was duly checked, similar plasmids expressing V5 epitope-flanked equine (NP_001075961), porcine (NP_999226), and bovine (NP_776365.1) Mx1 were synthesized. 

The nature and sequence of the V5-Mx1 proteins have previously been confirmed by resequencing the product of transgene-specific PCRs from transfected HEK-293T cell extracts [30]. Furthermore, the expression of recombinant V5-Mx1 proteins after transfection has been proven via the immunofluorescent detection of the V5 epitope in HEK-293T cells and immunoblotting [30]. It was hence confirmed that RFP-associated fluorescence is seen in the nucleus, whereas V5-Mx1 accumulates in the cytoplasm. In addition, the presence of each V5-Mx1 protein at the foreseen molecular weight was confirmed via immunoblotting [30]. The four expression plasmids were approximately the same size (between 7901 and 7946 bp), and the sequences of all the inserts of interest were duly verified via sequencing before use. All plasmid maps are available on request. 

#### 2.1.3. Antibodies 

The rabbit polyclonal anti-V5 conjugated to phycoerythrin used in the flow cytometry assays was from Abcam (catalog #ab72480), as was the monoclonal anti-V5 conjugated to FITC used in the immunofluorescence studies (#ab1274). The primary mouse monoclonal targeting viral NP was from the Cambridge Bio (#01-05-0145), and the corresponding secondary polyclonal goat anti-mouse IgG conjugated to FITC used in the flow cytometry assays was from Abcam (#ab6785). The HRP-conjugated anti-V5 tag and anti-actin monoclonals used in immunoblotting were from ThermoFisher, Waltham, MA, USA (MA5-15253) and Abcam, Boston, MA, USA (ab49900), respectively. 

#### 2.1.4. Virus 

The SBV stock was obtained through the amplification of the SBV-BH80/11-4 isolate in BHK-21 cells (ATCC^®^#CCL-10TM). 

### 2.2. Cell Expansion and Transfection 

Thawed HEK-293T cells (ATCC^®^CRL-1573TM) were first passaged once in DMEM (with 10% FCS, 1% pen-strep, and 0.5% fungizone) and then seeded onto 24-well plates (1 × 10^5^ cells per well), for flow cytometry. Eighteen hours later, cell monolayers were transfected with pDA657RA/V5-Mx1 using the Lipofectamine^®^3000 Transfection reagent kit (ref.#L3000-008, ThermoFisher). Briefly, in each well on a 24-well plate, 0.5 μg of plasmid DNA and 1 µL of P3000 reagent (1:2 *w*/*v* ratio) were diluted in 25 µL of Opti-MEM medium; the mix was then combined with 1 µL of Lipofectamine 3000 reagent diluted in 25 µL of Opti-MEM medium. The final mix was added dropwise to the cultured cells, which were reincubated at 37 °C for 24 h in a 5% CO_2_ and 80% RH atmosphere.

### 2.3. In Vitro Assay of Mx1 Anti-SBV Activity 

To evaluate the antiviral activity associated with the expression of each of the targeted V5-Mx1 proteins, the percentage of Schmallenberg virus nucleoprotein-positive cells was measured via flow cytometry in the infected populations of lipofectamine-transfected HEK-293T cells. Briefly, 24 h after transfection, cells were infected with the SBV stock at a multiplicity of infection of 0.5 to obtain an equal proportion of infected and uninfected cells in the same well. After 1 h of infection at 37 °C, the cells were rinsed with PBS, placed in fresh medium, and cultured for another 4 h. The duration of 4 h of virus amplification was decided after preliminary infections to find a balance between a strong virus signal and no secondary cell infection. The cells were then fixed with 4% paraformaldehyde, permeabilized in 0.2% saponin and 1% bovine serum albumin (BSA) in phosphate-buffered saline (PBS), and blocked with 1% BSA in PBS. A probing step was then carried out, using the corresponding primary antibodies to simultaneously detect the V5 epitope and the viral NP. The incubation conditions were 90 min at 4 °C. The solution contained the polyclonal anti-V5 at 0.1 µg/µL and the monoclonal anti-NP at a dilution of 1/500. A second incubation for 90 min at 4 °C was carried out using secondary polyclonal anti-mouse IgG at a dilution of 1/500.

Finally, the samples (>100,000 cells) were analyzed using an LSRFortessa flow cytometer (BD Biosciences, San Jose, CA, USA). Instrument settings were adjusted using fresh HEK-293T samples in 4 tubes (1 unlabeled sample, 2 singly labeled samples, and 1 sample labeled with the mixed antibodies) and the FACS Diva calculation software (BD Biosciences). The detection and compensation stabilities were tested before each experiment with SetUp beads, and dot plots were analyzed via the FACS DivaTM software (v.8.0.1, BD Biosciences). 

### 2.4. Data Analysis 

All values are presented as means ± SDs. Statistical analysis was performed via analysis of variance (ANOVA) with post hoc testing using Fisher’s protected least significant difference multiple range test. 

## 3. Results

### 3.1. Effect of V5-Mx1 Proteins on Schmallenberg Virus NP Synthesis

Given the reproducibility of the apparent MOI (% NP-positive cells) measured from V5-negative cells within each experimental campaign (gray boxes, Figure 1), it is assumed that the transfection procedure and reagents affected all the transfected/infected cell populations equally (Figure 1 and Figure 2). It is noteworthy that it was previously shown that after transfection with a control plasmid harboring a canine Mx1 inactivated by five successive stop codons, the infection rate of RFP-positive/V5-positive cells was similar to that of RFP-negative/V5-negative cells, which shows that the expression of RFP per se does not alter the viral biological cycle of the studied cells [17]. Among Mx1-negative cells, exposure to lipofectamine alone decreased SBV infectivity (*p* < 0.05), and exposure to any of the four lipofectamine/plasmid cocktails tested resulted in a further decrease (Figure 1). Interestingly, the viral NP detection rate among V5-negative cells was systematically lower when the cells were exposed to the cocktail-containing lipofectamine and the plasmid encoding the V5-btMx1 protein (Figure 1, *p* < 0.05). Furthermore, a systematic depletion (*p* < 0.001) of SBV NP-positive cells was detected in all V5-Mx1-expressing cell subpopulations, compared to the corresponding Mx1-negative subpopulations (Figure 1). This is attributable to the V5-Mx1 proteins themselves, thus suggesting a strong anti-SBV effect at the studied timepoint. 

### 3.2. Comparative Anti-SBV Activity among Different V5-Mx1 Proteins

Overall, the expression of the viral NP was significantly less hampered by canine V5-Mx1 than by any of the other three V5-Mx1 proteins (Figure 1, *p* < 0.05). When examining the distribution of fluorescence in the raw scatterplots, it can be intuitively perceived that the number of NP-positive cells decreases as the intensity of the V5-associated fluorescence increases (Figure 2). In order to quantify this trend, we arbitrarily created five consecutive V5-associated fluorescence ranges (Figure 3) and measured the viral NP expression rates characteristic of each (Figure 4). By doing so, we show that detection of the SBV NP five hours after infection in V5-Mx1-expressing cells is as diminished/delayed, as the level of expression of the recombinant V5-Mx1 protein targeted is high, suggesting an Mx1 concentration-dependent gradient in viral NP detection. Again, the canine Mx1-associated anti-Schmallenberg effect, though significant, appears weaker than those exercised by other Mx1 proteins. 

## 4. Discussion

The anti-SBV function of four orthologous Mx1 proteins localized in the cytoplasm was examined using a highly standardized experimental design. The common epitope (V5) grafted in the N-terminal position made it possible to use the same mAb to detect them. The fluorescence emitted by this antibody was used as a quantitative measure of the intracellular Mx1 concentration, and only experimental campaigns with a similar V5-specific mean fluorescence for the four V5-Mx1s being tested (CV < 10%) were retained for analysis. The results can therefore be interpreted in terms of the intrinsic antiviral activities of the recombinant Mx1 studied. Importantly, our results suggest that the four tested Mx1 proteins (originating from *Bos taurus*, *Canis familiaris*, *Equus caballus*, and *Sus scrofa*) exert anti-Schmallenberg virus activity.

The antiviral spectrum of the bovine Mx1 is probably the most studied after that of human MxA. Anti-rhabdovirus activity has been detected repeatedly in several experiments conducted in vitro, either against VSV [31,32,33] or against the rabies virus [34]. On the other hand, all the paramyxoviruses tested in vitro thus far (Sendai virus, bovine or human isolates of parainfluenza-3 virus, or bovine or human strains of respiratory syncytial virus) have been shown to be bovine Mx1-resistant [35]. In contrast, in vivo, transgenic mice expressing bovine Mx1 are much more resistant to the mouse pneumovirus than their wild-type counterparts [36]. In an experimental design almost comparable to that used here, it was established that the expression of an ectopic bovine Mx1 in HEK-293 cells caused a drastic decrease in the number of influenza A virus NP-positive cells [30]. Here, for the first time, we show that, similarly to the human MxA and the Mx2 proteins of mice, rats, and cotton rats, the bovine Mx1 protein exerts an inhibitory function on the life cycle of an orthobunyavirus, the Schmallenberg virus. As the natural infections of adult cattle with this virus remain almost asymptomatic, we suggest that the Mx1 protein participates in the innate control of the infection. On the other hand, in the bovine fetus, the immaturity of the IFN type 1 response and the resulting impairment of Mx1 expression may explain the unbridled proliferation of the virus and the associated developmental consequences [24]. 

The host range of orthobunyaviruses has been correlated to the inhibition efficiency of interferon-dependent effectors. For example, while human orthobunyaviruses (LACV and Oropouche virus) are inhibited by the ovine bone marrow stromal antigen 2 (BST-2), SBV is not, and the case is the opposite for human BST-2 [37]. Here, using Mx1 proteins from host and non-host species, such a correlation was not apparent.

Until recently, little information about the equine Mx1 protein was available. At most, we knew that the corresponding gene was located on chromosome 26 [38], that two spots were detected using antisera raised against human MxA on an electrophoresis gel of total proteins extracted from equine cells exposed to IFN-α [39], and that equine Mx1 exerted anti-influenza activity [30]. The antiviral activity of equine Mx1 against Thogoto virus was demonstrated recently [40]. The results gathered here confirm the role of equine Mx1 as an antiviral effector by showing that it also displays significant anti-Schmallenberg virus activity, the strength of which matches that of bovine Mx1 (Figure 1 and Figure 4). 

The porcine Mx1 gene and promoter were shown to share major structural and functional characteristics displayed by their homologs described in mice and humans [41]. In addition, allelic polymorphisms generating two very different isoforms have been identified [42,43]. The anti-influenza activity conferred by both V5-ssMx1 isoforms was evaluated in vitro using either the transfection of 3T3 cells followed by plaque assays [42] or transfection of Vero cells followed by the flow cytometric determination of the fraction of influenza virus-infected cells among Mx1-producing and nonproducing cell populations [43]. Both studies revealed that isoform α of ssMx1 is endowed with significant anti-influenza activity. This result was further consolidated by highlighting a blockade of the centripetal traffic of the incoming viral particles in the presence of ssMx1 [44]. In addition, porcine Mx1 and Mx2 proteins have recently been shown to inhibit the proliferation of several viruses in culture, i.e., pseudorabies virus [45], Senecavirus A virus [46], classical swine fever virus [47], and foot-and-mouth disease virus [48]. Our results are thus in line with previous studies reporting on the antiviral activities of porcine Mx1 and can add *Bunyavirales* to the growing list of its targets.

When canine Mx1 was transiently expressed in 3T3 cells, no anti-VSV activity was detected, which contrasts with canine Mx2 [49]. Similarly, using the minireplicon system for modeling influenza A and B viruses’ biological cycle, the expression of canine Mx1 did not break the polymerase activity at all [50,51]. In a recent study, the influenza A virus NP-positive cell fraction did not vary significantly in the presence of canine Mx1, suggesting, as in previous studies, that it is not endowed with anti-influenza activity [30]. Here, we show that canine Mx1 decreases or delays the synthesis of the SBV NP, with a weaker effect than that demonstrated by the porcine, bovine, and equine Mx1 proteins but nevertheless significant (*p* < 0.05). To our knowledge, this is the first time that it has been shown that the life cycle of a virus is inhibited by this canine Mx1. In summary, all Mx1 proteins tested here exert an apparent anti-Schmallenberg virus activity. Since the experimental design was highly standardized and the expression levels of ectopic Mx1 were similar, both with respect to the transfected cell fraction and the average fluorescence intensity emitted by the anti-V5 mAb, the amplitude of the measured anti-SBV effects can be compared. In doing so, it appears (i) that the anti-SBV effect is approximately similar for bovine, equine, and porcine Mx1, and (ii) that canine Mx1 exerts an antiviral activity that is slightly weaker than that shown by its three orthologs (Figure 1). 

This study showed that the fraction of NP-positive cells decreased steadily as the intensity of the V5-associated fluorescence increased for all Mx1 proteins tested (Figure 4). In other words, the more a cell expresses a given ectopic Mx1 protein, the more unlikely the viral NP will be detectable 5 h pi. To our knowledge, the highly standardized design of this study allows us to interpret this result in terms of a dose–response relationship between the cytoplasmic concentration of the ectopic Mx1 protein on one hand, and the rate of viral neo-NP synthesis in the same cytoplasm on the other hand. We also showed that, beyond a certain concentration threshold, Mx1 completely abolished the synthesis of the viral NP (Figure 4). This result suggests that, in nature, it is not only the amino acid sequence of a given Mx1 protein that can make the difference in terms of innate immunity but also (especially) the magnitude of its expression—hence the intrinsic nature of the relevant gene promoter. It should, however, be noted that the used readout only measures a partial run of the viral biological cycle, since the steps downstream of NP synthesis do not affect the results. Therefore, it is quite possible that certain Mx1 proteins may also differently brake other viral proteins’ synthesis, viral genome replication, assembly, and/or budding processes. 

The human MxA protein is known to inhibit various *Bunyaviruses*: LACV (genus *Orthobunyavirus*); CCHFV and Dugbe virus (*Nairovirus*); RVFV (*Phlebovirus*); and HTNV, PUUV, and TULV (*Hantavirus*) [14,15,16,17,18]. This does not mean that it has antiviral activity against all members of the *Peribunyaviridae* family. For example, viruses of the genus *Tospovirus* and *Bunyamweravirus*, the archetype of the genus *Orthobunyavirus*, escape the MxA blockade [15,52]. Here, we show that the life cycle of another member of the genus *Orthobunyavirus*, SBV, is susceptible to the four tested mammalian isoforms of Mx1.

In terms of the underlying mechanism, it is the interaction between MxA and LACV that is by far the most studied thus far [16,18,19,53]. Similar to other members of the *Peribunyaviridae* family, the genome of LACV consists of three single-stranded RNA segments of negative polarity. After entry into the host cell, LACV transcribes and replicates its genome in the cytoplasm. The genomic RNA segments encode three structural proteins: the RNA polymerase, a glycoprotein precursor (which is processed into the envelope glycoproteins G1 and G2), the nucleocapsid protein (N), and two nonstructural proteins (NSm and NSs). The genomic RNAs of the incoming virus are first transcribed into mRNA by the viral RNA polymerase. Following mRNA translation, the replication of the genome occurs via the synthesis of full-length plus-strand copies of the genomic RNA. For this shift from mRNA transcription to genome replication, a newly synthesized N protein is required [54]. Presumably, the unassembled N protein associates with the viral polymerase to form the replication complex and tightly associates with the vRNAs to form the viral ribonucleoprotein complexes (vRNPs). These complexes are transported to the Golgi apparatus, where they associate with the viral envelope glycoproteins and bud into the Golgi cisternae, thus releasing new virions [55]. 

With respect to the molecular mechanism underlying inhibition of LACV by MxA, the literature data are not unambiguous. MxA was shown to inhibit LACV replication by sequestering the newly produced viral NP at large, including membrane-associated perinuclear complexes, with this mis-sorting preventing the NP from performing its function in viral replication [18,53]. In contrast, Frese and colleagues [16] suggested that the active viral RNA polymerase complex is the target for MxA. They demonstrated that the human MxA protein inhibits the accumulation of viral transcripts, particularly the longer ones, suggesting an effect on elongation [16]. Altogether, it seems that this effect on elongation could also be secondary to NP sequestration. This sequestration would probably also affect the large segments, as they need more of the nucleocapsid protein for their replication. Some results suggest a similar mode of action against other viruses of this family, including CCHFV, RVFV, HTNV, PUUV, and TULV [14,16,17,56]. However, an argument against a general anti-bunyaviral mechanism is the observation that the NP of Dugbe virus appears to avoid MxA activity and is not sequestered to the perinuclear region by human MxA [15]. 

Here, the average intensity of NP-associated fluorescences did not decrease when the average intensity of V5Mx1-associated fluorescences increased (Figure 2). This observation excludes the intervention of a competition between Mx1 and mAb # 01-05-0145 (Cambridge Bio) for the same binding element or epitope on the NP. Thus, the Mx1 proteins tested here caused a drastic dose-dependent decrease in the number of cells expressing a detectable amount of the NP 5 h pi. At the timepoint of 5 h pi, a time when secondary infections are excluded, the experiment does not inform about a possible effect of Mx1 on the replication, assembly, and/or budding of new viral particles. On the other hand, it tests the event chain, moving from the binding to the synthesis of the NP, while passing by the primary transcription of the genomic segment “S” during the first viral cycle. Our study thus demonstrates that, under the conditions of the experiment, the four tested Mx1s hinder one or more of the early stages of the viral cycle. This result is compatible with the blocking of the primary transcription, as observed previously for LACV by Frese and colleagues [16], but does not exclude that, downstream of the synthesis of the NP, the tested Mx1s have the same anti-replicative function as that shown when testing MxA/LACV, MxA/HTNV, and MxA/RVFV pairs. 

The abrupt decrease in the number of NP-positive cells observed in fraction 4 suggests a radical change in the inhibitory mechanism. In the low-concentration range, the inhibition appears to be of a competitive nature, whereas blocking occurs at higher concentrations. As the most effective way to prevent viral replication is to block the entry of the parental virus, it is suggested that, at high concentrations, the Mx1 proteins inhibit this early stage of the viral cycle. Knowing that orthobunyaviruses enter cells via a clathrin-dependent mechanism [57] and knowing the structural proximity between clathrins and Mx proteins [58], it would be interesting to investigate whether clathrin-dependent endocytosis is impaired at high concentrations of Mx proteins.

## Figures and Tables

**Figure 1 viruses-15-01055-f001:**
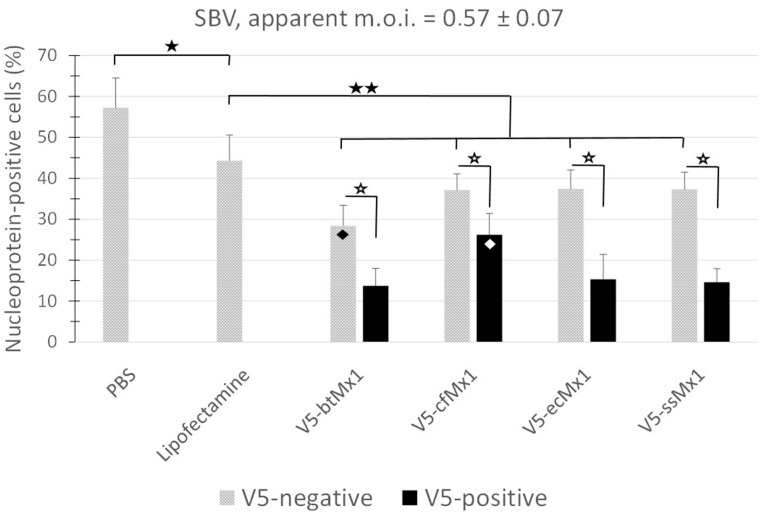
Percentage of Schmallenberg virus nucleoprotein-producing cells 5 h after Schmallenberg virus infection in HEK-293T cell populations. Each box pair represents means ± SDs measured in V5-Mx1-nonexpressing (gray) and expressing (black) cells from four independent experiments, each including three different wells. The expression of bovine (bt), canine (cf), equine (ec), and porcine (ss) V5-Mx1 caused the significant inhibition of viral nucleoprotein expression (hollow stars, *p* < 0.001). Note that the exposure to lipofectamine or to a lipofectamine/plasmid complex significantly decreased NP detection in Mx-negative cells (solid stars, one: *p* < 0.05 and two: *p* < 0.01, respectively). The lozenges indicate statistical differences (*p* < 0.01) between expression plasmids: (i) the rate of NP expression among Mx1-negative cells was lower when the plasmid encoded V5-btMx1 rather than any other Mx1 (black lozenge), and (ii) the rate of NP expression in Mx1-positive cells remained higher when the plasmid encoded V5-cfMx1 rather than any other Mx1 (white).

**Figure 2 viruses-15-01055-f002:**
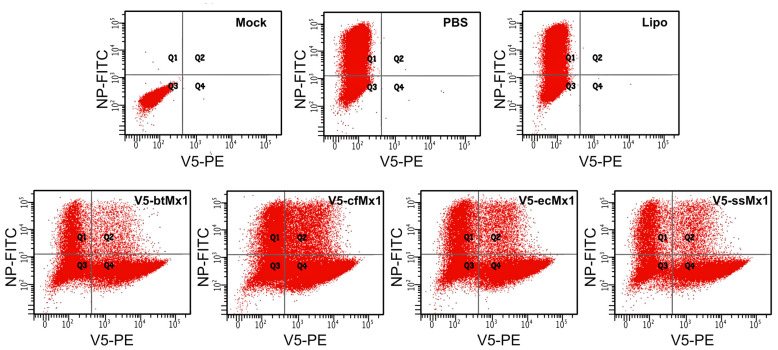
Schmallenberg virus nucleoprotein synthesis was inhibited or delayed in the presence of specific Mx1 proteins. Typical raw data obtained after the flow cytometric analysis of double-stained HEK-293T cells that were transfected/infected or not. The expression of exogeneous V5-Mx1 proteins was driven by the corresponding plasmid transiently transfected 24 h before infection, with each transfection process thus resulting in a mixed population of expressing (V5 epitope-positive) and non-expressing (V5 epitope-negative) cells. Where appropriate, cell populations were infected with Schmallenberg virus 5 h before analysis. Cells were double-immunostained, as described in the text, with FITC detecting viral nucleoprotein and phycoerythrin (PE) detecting the Mx1 V5-epitope. Cell suspensions were analyzed using a LSRFortessa flow cytometer, gating on forward/side scatter to exclude debris. A minimum of 10^5^ cells were acquired and analyzed with BD-FACSDiva software v.8.0.1.

**Figure 3 viruses-15-01055-f003:**
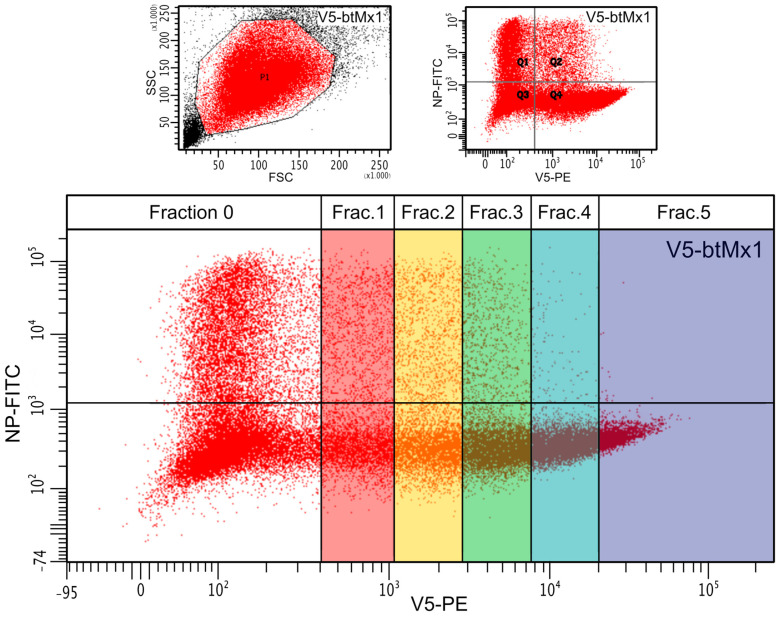
Schematic description of the categorization of Mx1-positive cells as a function of the intensity of the fluorescence associated with the V5 epitope and, therefore, as a function of the target recombinant Mx1 level of expression. The denomination of the five consecutive fluorescence ranges corresponds to that used in Figure 4. See Figure 2 for the key.

**Figure 4 viruses-15-01055-f004:**
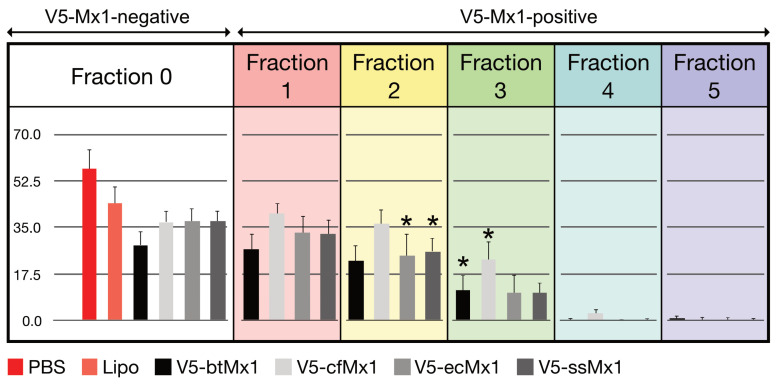
The synthesis of Schmallenberg virus nucleoprotein 5 h after infection in Mx-expressing cells is as inhibited or delayed as the level of expression of the recombinant Mx1 protein targeted is high. Note that the canine Mx1-associated anti-Schmallenberg effect, though significant, is clearly weaker than that exercised by other Mx1 proteins. Each box represents the mean ± SD of the fractions of viral nucleoprotein-positive cells characterizing mock- (black), lipofectamine- (gray), or lipofectamine/plasmid-pretreated (colors) HEK-293T cells appearing in the respective V5-associated fluorescence categories defined in Figure 3. For each pretreatment (PBS, lipofectamine, or lipofectamine/plasmid), the means ± SDs were calculated from twelve independent wells obtained from four experimental campaigns organized at least 1 week apart (three wells per campaign). See Figure 3 for the key. For said Mx1, * represents the first fraction in which the rate of NP expression was significantly lower than that among Mx1-negative cells (*p* < 0.05).

## Data Availability

The data presented in this study are available in the article and are available on reasonable request from the corresponding author.

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
