# Peer review of "Anti-Schmallenberg Virus Activities of Type I/III Interferons-Induced Mx1 GTPases from Different Mammalian Species"

_viruses, 2023, doi:10.3390/v15051055_

Round 1

Reviewer 1 Report

The present study describes the antiviral activity of of type I/III interferons-induced Mx1 GTPases against the bunyavirus Schmallenberg. The topic is very interesting and the results clearly exposed. 

Minor comments:

1. Why did the author select this virus? This detail is missing;

2. Are there any other studies on Mx1 proteins in the Bunyaviridae family?

3. Deepen the SBV growth in cell culture. This information is missing in Materials and methods section;

4. Is the suddivision in paragraphs necessary in the Discussion?

Reviewer 2 Report

In this manuscript titled “Anti-Schmallenberg virus activities of type I/III interferons-induced Mx1 GTPases from different mammalian species”, by comparing the number of viral nucleoprotein positive cells 5 hours after infection in a sample of 100,000 cells expected to contain both Mx1 positive and Mx1-negative cell subpopulations, the authors measured the anti-Schmallenberg virus activity of Mx1 proteins in four different mammals. And it is proved that this antiviral activity is a dose-dependent phenomenon. The authors conclude that the four mammalian Mx1 isoforms have strong anti-Schmallenerg activity. The manuscript is well-organized and has certain significance. However, there are still a lot of problems in this manuscript that need to be revised. I would suggest accepting it after the following concerns are addressed.

1.        The language needs considerable attention.

2.        The analysis method is simple and needs more innovation.

3.        The quality of pictures needs to be optimized.

4.        The abstract contains too much information and does not summarize the meaning of the full text. Try to be concise; The content in the introduction should not be consistent with the abstract, lines 39-41 and 81-91; please explain the significance of this manuscript in the abstract and introduction.

5.        Type III interferon is IFN-γ, not IFN-l, in line 39ï¼›Lines 45-48 and 69-70 need references to support them.

6.        The title "Biochemical and biologic reagents" in line 119 can be omitted, and the contents in lines 194-196 can be omitted. Lines 438~439 should be deleted.

7.        Figure 1 should indicate the meaning of solid and hollow five-pointed stars.

8.        The references are old and need more new references.

9.        There are many non-standard expressions in the article. For example, "p<0.05" needs to be changed to "p<0.05"; the abbreviations of proper nouns should be consistent. For example, "mx1" in line 52 and "Mx1" in line 74 are different.

10.     Please pay attention to the writing format problem. Such as, hyphenated spaces between words on lines 44, 50, 74, 123, 143,178 and 199; "4" in line 275 and "four" in line 277; the "," need to be changed ";" in line 34; "canine" in line 31 and "dog" in line 331; "a m.o.i. of 0,5" in line 115, "CO2" in line 122, "ml" in line 124, "0,5% fungizone" in line 162, "µl" in lines 166-167, "80%RH" in line 169, "0,2% saponin" in line 178, and "per se" in line 204; in Line 49, the abbreviation "A2G" needs a full name; punctuation should be unified, such as line 34; there is only one sentence in lines 367-368, which is unsuitable as a separate paragraph. This sentence should be put in other suitable places.

11.     Please correct the format and sentences in lines 141~142, "I was hence confirmed that RFP- associated fluorescence is seen in the nucleus whereas V5-Mx1 accumulates in the cytoplasm.", so as to avoid the appearance of the first person.

12.     In the discussion part, the author's opinions are appropriately added to stimulate readers' interest in the research prospect.

13.     Just listing the results is boring. The author can add some analysis or discussion about those results in the abstract to increase the study's novelty.
